# Subclass Analysis of Malignant, Inflammatory and Degenerative Pathologies Based on Multiple Timepoint FAPI-PET Acquisitions Using FAPI-02, FAPI-46 and FAPI-74

**DOI:** 10.3390/cancers14215301

**Published:** 2022-10-28

**Authors:** Frederik M. Glatting, Jorge Hoppner, Hans-Ulrich Kauczor, Peter E. Huber, Clemens Kratochwil, Frederik L. Giesel, Uwe Haberkorn, Manuel Röhrich

**Affiliations:** 1Department of Nuclear Medicine, University Hospital Heidelberg, 69120 Heidelberg, Germany; 2Clinical Cooperation Unit Molecular and Radiation Oncology, German Cancer Research Center (DKFZ), 69120 Heidelberg, Germany; 3Department of Radiation Oncology, University Hospital Heidelberg, 69120 Heidelberg, Germany; 4Department of Diagnostic and Interventional Radiology, University of Heidelberg, 69120 Heidelberg, Germany; 5Translational Lung Research Center Heidelberg (TLRC), Member of the German Center for Lung Research DZL, 69120 Heidelberg, Germany; 6Department of Nuclear Medicine, Heinrich-Heine-University, Medical Faculty, University Hospital, Duesseldorf, 40225 Duesseldorf, Germany; 7Clinical Cooperation Unit Nuclear Medicine, German Cancer Research Center (DKFZ), 69120 Heidelberg, Germany

**Keywords:** fibroblast activation protein, FAPI, PET, cancer, inflammation, degeneration, pancreatic carcinoma, pancreatitis

## Abstract

**Simple Summary:**

In a recent retrospective analysis, we evaluated the diagnostic value of repetitive early FAPI-PET-imaging with FAPI-02, FAPI-46 and FAPI-74 for malignant, inflammatory/reactive and degenerative pathologies in 24 cancer patients. Here, we apply a subgroup analysis to that dataset. Differential uptake behavior over time was observed in several subclasses of malignant lesions, inflammatory/reactive lesions and degenerative lesions. These differences over time were particularly manifested in the direct comparison between the uptakes associated with pancreatic carcinoma (stable or increasing over time) and inflammatory lesions of the pancreas (markedly decreasing over time). We conclude that multiple timepoint FAPI-PET/CT is a promising innovative imaging technique that provides additional imaging information compared to single timepoint imaging.

**Abstract:**

Purpose: FAPI-PET is a promising imaging technique for various malignant as well as non-malignant pathologies. In a recent retrospective analysis, we evaluated the diagnostic value of repetitive early FAPI-PET-imaging with FAPI-02, FAPI-46 and FAPI-74 for malignant, inflammatory/reactive and degenerative pathologies. Here, we apply a subgroup analysis to that dataset and describe the tracer-wise uptake kinetic behavior of multiple types of FAPI-positive lesions, which are encountered frequently during clinical routine. Methods: A total of 24 cancer patients underwent whole-body FAPI-PET scans, and images were acquired at 10, 22, 34, 46 and 58 min after the administration of 150–250 MBq of ^68^Ga-FAPI tracer molecules (eight patients each regarding FAPI-02, FAPI-46 and FAPI-74). Standardized uptake values (SUVmax and SUVmean) of healthy tissues, cancer manifestations and non-malignant lesions were measured and target-to-background ratios (TBR) versus blood and fat were calculated for all acquisition timepoints. Results: Differential uptake behavior over time was observed in several subclasses of malignant lesions, inflammatory/reactive lesions and degenerative lesions. These differences over time were particularly manifested in the direct comparison between the uptakes associated with pancreatic carcinoma (stable or increasing over time) and inflammatory lesions of the pancreas (markedly decreasing over time). Furthermore, marked differences were found between the three tracer variants regarding their time-dependent uptake and TBRs within different subclasses of malignant, inflammatory/reactive and degenerative pathologies. Conclusion: Multiple timepoint FAPI-PET/CT is a promising innovative imaging technique that provides additional imaging information compared to single timepoint imaging. Differences in the kinetic behavior of malignant and benign pathologies can facilitate the interpretation of FAPI-positive lesions.

## 1. Introduction

Numerous studies have demonstrated the diagnostic potential of positron emission tomography with inhibitors of fibroblast activation protein (FAPI-PET). FAPI-PET was introduced as a new imaging method for tumors targeting FAP-positive cancer-associated fibroblasts, which are present in a large variety of cancers [1,2,3,4]. However, multiple other pathologies, such as fibrotic diseases [5,6], degenerative processes [7], inflammatory processes, [8,9] and reactive tissue [10] contain FAP-positive fibroblasts. According to that, many non-malignant pathologies, such as IgG4-related disease, pulmonary and renal fibrosis, liver cirrhosis and arthritis, show marked tracer uptake in FAPI-PET [11,12,13,14,15]. The differentiation of malignant from non-malignant FAPI-PET positive lesions still makes the interpretation of FAPI-PET scans challenging [16,17]. To date, only few studies have analyzed potential differences of FAPI-uptake between malignant and non-malignant lesions [11,17,18]. Subclasses of malignant lesions (primary tumors, local recurrences, and different forms of metastases) have been analyzed for single time point FAPI-PET imaging (e.g., [19,20]), but potential differences in FAPI-PET of subclasses of non-malignant tissues have not yet been described.

In several publications so far, it has been suggested that dual or multiple timepoint imaging may increase the diagnostic accuracy of FAPI-PET [10,11,18]. Thus, in a recently published retrospective analysis, we evaluated a repetitive early FAPI-PET imaging protocol with PET acquisition at 10, 22, 34, 46 and 58 min after tracer application, wherein three FAPI-tracer variants (FAPI-02, FAPI-46 and FAPI-74) were applied in eight patients, respectively [21]. We found marked differences of the three tracers regarding their biodistribution and their kinetic behavior in malignant, degenerative and inflammatory as well as reactive lesions. However, an analysis of different subgroups of malignant lesions (e.g., primary versus local recurrence or metastases at different localizations) or benign lesions (e.g., inflammatory lesions of the pancreas (IPa) versus reactive tissue (R) or postoperative lesions (RPo) has not yet been performed. Here, we apply a subgroup analysis to the dataset of our prior analysis [20] in order to characterize the uptake kinetic behavior of relevant subclasses of malignant and non-malignant FAPI-positive lesions.

## 2. Materials and Methods

### 2.1. Patient Characterization

The clinical characteristics of the patients of this dataset have already been described in our previous analysis [21]. Briefly, 24 cancer patients (aged 34 to 83 years, average 61 years) without surgery, radiotherapy or chemotherapy within the last 4 weeks underwent repetitive early ^68^Ga-FAPI-PET/CT. Median intervals between treatments and ^68^Ga-FAPI-PET imaging were 13 months (range 1 to 240) for surgery, 29 months (range 2 to 260) for radiotherapy and 7 months (range 1 to 240) for chemotherapy. All patients were referred by their treating physicians for ^68^Ga-FAPI-PET/CT due to clinical indications. All patients signed informed consent and agreed to the scientific evaluation of their personal data. This retrospective analysis was approved by the local institutional review board (study number S-115/2020).

### 2.2. Repetitive FAPI-PET/CT Imaging

Diagnostic imaging was performed under the conditions of the updated Declaration of Helsinki, § 37 (unproven interventions in clinical practice) and in accordance with German Pharmaceuticals Law §13 (2b) for medical reasons. FAPI tracers (FAPI-02, FAPI-46 and FAPI-74 with eight patients each) were labelled with ^68^Ga as previously described [22,23] and applied intravenously (80 nmol/GBq). CT scans were performed within the first 10 min after tracer injection with a Biograph mCT Flow™ PET/CT-Scanner (Siemens Medical Solutions) using the following parameters: slice thickness of 5 mm, increment of 3–4 mm, soft-tissue reconstruction kernel, and care dose. PET scans were acquired exactly 10, 22, 34, 46 and 58 min post tracer administration (timepoints 1, 2, 3, 4 and 5) with a standardized field of view allowing whole-body scans within 12 min in 3D (matrix 200 × 200) in FlowMotion™ with 1.6 cm/min. Emission data were corrected for random, scatter and decay. Reconstruction was conducted with an ordered subset expectation maximization (OSEM) algorithm with 2 iterations/21 subsets and Gauss-filtered to a transaxial resolution of 5 mm at full-width half-maximum. Attenuation correction was applied based on low-dose non-enhanced CT data.

### 2.3. Image Analysis and Quantification

Quantitative assessment of standardized uptake values (SUV) using a volume of interest (VOI) technique was carried out by FMG and MR independently and finalized by consensus. Tracer biodistribution in patients and uptake of pathologies were quantified by mean and maximum SUVs (SUVmean and SUVmax). Normal organs and pathologies were contoured and analyzed using PMOD software as described before [21]. All VOIs for normal tissues and pathologies were defined within the FAPI-PET images acquired at 58 min post injection (p.i.), using isocontour of 50–70%, and automatically transferred to the previous timepoints by PMOD software in order to achieve identical intra-individual VOIs. Classes and subclasses of malignant, inflammatory/reactive or degenerative lesions were determined based on clinical information and CT-morphology. Only lesions with highly suggestive characteristics for one of the (sub-)classes were included. Increase, constant level or decrease of ^68^Ga-FAPI-uptake were determined based on visual assessment and the trends of the mean absolute values of SUVmax or SUVmean over time.

### 2.4. Statistical Analysis

We performed descriptive analyses for patients and their characteristics. For determination of SUVs, median and range were used. The correlation of FAPI-uptake within or outside the lesions was determined by using a two-sided *t*-test. A *p*-value of <0.05 was defined as statistically significant. Excel version 2111 and Origin2021b were used for statistical analyses.

## 3. Results

### 3.1. Tracer Uptake and TBRs of Malignant Manifestations over Time

Table 1 summarizes the number of analyzed malignant lesions, including primaries, local recurrences and metastases at different localizations. The corresponding uptake kinetics in terms of SUVmax and SUVmean is illustrated in Figure 1; the corresponding TBR_SUVmax_ and TBR_SUVmean_ versus blood and fat tissue over time in the Appendix A.

Generally, a higher uptake as well as TBRs both versus blood and versus fat tissue were observed for FAPI-46 in comparison with FAPI-02 and FAPI-74.

Respecting FAPI-02, local recurrences and lymphogenic metastases showed a positive slope for uptake in terms of SUVmax and SUVmean, whereas the other subclasses featured an approximately constant uptake over time. Moreover, the corresponding TBRs versus blood increased over time for all hereby considered subclasses of malignancies, whereas TBRs versus fat tissue remained roughly constant for all subclasses apart from local recurrences and lymphogenic metastases showing a slight increase over time.

In contrast, FAPI-46 showed a slight decrease of SUVmax values for metastases, particularly for distant lesions over time, and approximately constant values for primaries and local recurrences, whereas according to the SUVmean values of malignant lesions, all metastases, most pronounced lymphogenic metastases, tended to increase over time. TBR versus blood showed an increasing curve progression for all malignant subclasses apart from TBR_SUVmax_ for distant metastases showing a constant level, whereas the calculation of TBR versus fat tissue led to constant or even slightly reducing TBRs over time, especially for TBR_SUVmax_ regarding distant and thus total metastases. Additionally, primaries featured lower absolute values of SUVmean, SUVmax, TBRs versus blood and fat tissue compared to the other classes of malignant lesions.

Concerning FAPI-74, there was a decreasing uptake in terms of SUVmax and SUVmean over time for primaries and local relapses in contrast to metastases manifesting an increase for all subclasses, which led to a constant uptake of the pooled malignant manifestations over time. Notably, the tracer uptake as well as TBRs versus blood and fat tissue were higher in primaries and recurrences than in metastases. The TBR_SUVmax_ as well as TBR_SUVmean_ versus blood increased over time for all subclasses, whereupon the TBR_SUVmax_ as well as TBR_SUVmean_ versus fat tissue showed decreasing levels over time for primaries and local recurrences but relatively constant levels for all types of metastases.

### 3.2. Tracer Uptake and TBRs of Metastases at Different Localizations over Time

The kinetics of the different types of metastases is summarized in Figure 2 and Appendix A as well as Appendix A, based on the listed lesions within Table 1.

Respecting FAPI-02, the uptake in terms of SUVmean and SUVmax for metastases is characterized by an increase over time for all subclasses apart from the osseous metastases, which started at a higher initial uptake level relative to the others and showed a decreasing uptake over time. These opposite trends led to a constant level of uptake for the pooled metastases. The TBRs versus blood featured an increase over time for all metastases, even for the osseous, but at different slopes. However, the TBR versus fat tissue resulted in a constant level, showing a dip at the third acquisition timepoint for all metastasis subclasses considering TBR_SUVmean_, apart from the decrease observed for osseous metastases.

In addition, for FAPI-46, the initial uptake of the single osseous metastasis is many times higher relative to the other analyzed metastasis manifestations. The SUVmax values remained constant over time for all metastasis subclasses, whereas the SUV mean values increased slightly over time for all subclasses apart from a constant uptake for the single osseous metastasis. The TBRs versus blood showed an increase over time at different slopes for all metastasis localizations. However, the TBR versus fat stayed roughly constant apart from the markedly decreasing TBRs of the solitary osseous metastasis over time.

Unlike the two other tracer variants, for FAPI-74 the uptake within osseous metastases in terms of SUVmean and SUVmax increased over time, similar to the pooled metastases and lymphogenic metastases but in contrast with the trend of hepatic metastases, showing a slight decrease over time. The initial uptake level of lymphogenic and hepatic metastases was higher than in the osseous metastases and stayed markedly higher over time. FAPI-74 showed increasing TBRs versus blood over time for all metastasis subclasses, whereas the TBR versus fat tissue featured the same trends as the tracer uptake in terms of SUVmean and SUVmax.

### 3.3. Tracer Uptake and TBRs of Inflammatory/Reactive Lesions over Time

Several subclasses of inflammatory and reactive pathologies are depicted in Table 2, Figure 3 and the Appendix A. The uptake kinetics of other specific lesions are not shown separately in the figures as they occur rarely, not for all tracer variants in the dataset, or did not match the other types of classifications are listed in Appendix A.

For FAPI-02, the SUVmax values decreased over time regarding the inflammatory lesions, whereas they showed an insignificant tendency over time for reactive lesions. In contrast with this, a slight increase of uptake in terms of SUVmean was observed for the pooled inflammatory/reactive lesions, which is made up by a constant uptake level of the pancreatitis-related inflammatory lesions and the increasing uptake of the postoperative reactive lesions. Thus, the TBRs versus blood increased over time for all types of inflammatory and reactive lesions but at different slopes. However, the TBRs versus fat decreased over time for the inflammatory lesions and remained approximately constant over time for the reactive lesions, resulting in slightly decreasing TBRs over time for the pooled inflammatory/reactive lesions.

Respecting FAPI-46, the pooled inflammatory and reactive lesions showed a constant level of uptake (SUVmax) or a slightly increasing uptake (SUVmean) over time. These trends were made up by decreasing SUVmax values and more or less constant SUVmean values of general inflammatory and pancreatitis-associated lesions in combination with increasing uptake values in terms of SUVmax and SUVmean concerning reactive, particularly postoperative, lesions. A similar trend of increasing TBRs versus blood was observed for all inflammatory and reactive lesions but at different slopes, whereas the calculation of TBRs versus fat led to differential behavior over time for inflammatory and reactive pathologies, as inflammatory lesions featured decreasing values (TBR_SUVmean,_ TBR_SUVmax_), whereas reactive lesions showed increasing values (TBR_SUVmean_) and an approximately constant level (TBR_SUVmax_).

FAPI-74 showed an increase of uptake in terms of both SUVmax and SUVmean for reactive lesions and a slight decrease for inflammatory lesions, resulting in an approximately constant level of uptake for the pooled inflammatory/reactive pathologies over time. Again, the TBRs versus blood increased over time for all subclasses at different slopes, whereas the TBRs versus fat tissue manifested differential time-dependent behavior between inflammatory and reactive lesions. Inflammatory lesions featured a decrease in TBRs versus fat values, whereas an increase of TBRs versus fat values were observed for reactive lesions.

### 3.4. Comparison of Physiological Pancreatic FAPI-Uptake, Pancreatitis- and Pancreatic Carcinoma-Associated FAPI-Uptakes and TBRs over Time

The time-dependent uptake and TBRs versus blood and fat tissue of pancreatitis-associated and pancreatic carcinoma-related tracer uptake are compared within Figure 4 and Appendix A for the three tracer variants. Table 3 depicts the number of analyzed physiological and pathological pancreatic tissues.

Regarding the principal biodistributional behavior of the physiological pancreas, the three tracer variants showed a decreasing uptake in terms of SUVmax and SUVmean over time [21]. Notably, FAPI-46 featured a decline of the TBRs versus blood over time indicating a faster decrease of uptake in physiological pancreatic tissue relative to the blood pool, while the TBRs versus blood remained constant for FAPI-02 and FAPI-74. A decrease of TBRs versus fat tissue over time was observed within healthy pancreatic tissue for all three tracer variants.

Regarding FAPI-02, the uptake within malignant lesions of pancreas increased more strongly over time compared with the pooled malignancies. However, the pooled inflammatory/reactive lesions featured a slight decrease for SUVmax and an approximately constant uptake level over time for SUVmean. The pancreatitis-associated uptake behaved in the same way, but the observed decrease of SUVmax was more pronounced than for the pooled inflammatory/reactive pathologies. The TBRs versus blood increased over time for all analyzed pathological subclasses but at different slopes; the highest for malignant and the lowest for inflammatory lesions. Notably, the pancreatic carcinoma-associated TBRs versus fat tissue showed an increase over time for both TBR_SUVmax_ and TBR_SUVmean_ in contrast with all the other pathological classes analyzed, which featured constant or declining values.

Noteworthy for FAPI-02, no significant difference in the absolute values of uptake could be observed between the inflammatory lesions of the pancreas compared with the pooled inflammatory lesions.

FAPI-46 showed a constant level of SUVmax over time for all pathological subclasses apart from the pancreatitis-related lesions indicating a slight decrease. The SUVmean increased over time for all pathological classes apart from the pancreatitis-related lesions featuring a plateau. The TBRs versus blood for both TBR_SUVmax_ and TBR_SUVmean_ increased over time for all pathological subclasses at different slopes, whereas the TBRs versus fat tissue for both TBR_SUVmax_ and TBR_SUVmean_ remained roughly constant over time for all pathological subclasses apart from the inflammatory pancreatitis-related lesions featuring a slight decline over time.

Notably for FAPI-46, there was no significant difference in the absolute values of uptake between the malignant as well as inflammatory lesions compared with the pooled malignant and inflammatory lesions.

For FAPI-74, a higher uptake in terms of SUVmax and SUVmean in pancreatic lesions, both in malignant and inflammatory types, was observed relative to the corresponding pooled malignant and inflammatory lesions. The pancreatic malignant and inflammatory lesions showed a high initial uptake relative to the corresponding pooled lesions. An approximately constant uptake was found for malignant and a slight decrease for inflammatory lesions of the pancreas over time, whereas the pooled lesions of malignant and inflammatory nature stayed approximately constant over time regarding SUVmax and SUVmean. The TBRs versus blood increased over time for all pathological classes considered, whereas the TBRs versus fat tissue followed the trends of the general uptake kinetics of SUVmax and SUVmean.

Figure 5 shows exemplary images of three patients with both pancreatic ductal adenocarcinoma (PDAC) manifestations and inflammatory lesions of the pancreas (ILP) examined by multiple timepoint ^68^Ga-FAPI-PET using FAPI-02, FAPI-46 and FAPI-74. In all three cases, mostly stable uptake of PDACs and markedly decreasing uptake of ILPs is visible.

### 3.5. Tracer Uptake and TBRs of Degenerative Lesions over Time

Furthermore, the numbers of degenerative lesions, including several subclasses, are summarized within Table 4, and the most relevant subclasses and the pooled lesions are visualized in Figure 6 as well as in Appendix A. Again, the uptake kinetics of other specific lesions that are not shown separately in the figures are listed in Appendix A.

Respecting FAPI-02, the pooled degenerative lesions as well as all of the analyzed subclasses featured an increase over time in terms of SUVmax and SUVmean apart from those at zygapophysial joints showing a constant level of uptake. The degenerative subclasses showed similar absolute values of uptake. Moreover, the TBRs versus blood, both TBR_SUVmax_ and TBR_SUVmean_, increased over time for all degenerative subclasses, whereas the TBRs versus fat remained approximately constant over time apart from the TBR_SUVmax_ as well TBR_SUVmean_ for the single degenerative lesion at the glenohumeral joint, featuring an increase over time.

For FAPI-46, the SUVmax kinetics showed an approximately constant uptake over time for all degenerative lesions apart from those at the glenohumeral joint and osteophytes, featuring a slight increase over time, whereas the SUVmean values increased over time for all classes of degenerative lesions at different slopes except the degenerative lesions at the temporomandibular joint and the insertion-related tendinopathy. Notably, the absolute values of uptake at osteophytes were markedly higher than those of the other classes of analyzed degenerative lesions. Again, TBRs versus blood (TBR_SUVmax_, TBR_SUVmean_) increased over time at different slopes for all types of degenerative lesions, whereas the TBRs versus fat tissue showed a more heterogeneous kinetics behavior in way that TBR_SUVmax_ remained roughly constant over time apart from degenerative lesions at the temporomandibular joint and the insertion-related tendinopathy and that TBR_SUVmean_ showed a constant level over time apart from the osteophytes and degenerative lesions at temporomandibular joint, both manifesting a slight decline over time.

FAPI-74 showed increasing absolute uptake values in terms of SUVmax and SUVmean over time for all types of degenerative pathologies analyzed and ergo the pooled degenerative lesions at different slopes, lowest for degenerative lesions located at acromioclavicular joints or sternoclavicular joints, whereupon all of the different types of lesions manifested a highly similar level of uptake in absolute values. Thus, an increase of TBRs versus blood (TBR_SUVmax_, TBR_SUVmean_) was observed for all degenerative classes over time with different gradients, except the degenerative lesions located at the acromioclavicular joint or sternoclavicular joint, manifesting an approximately constant level of TBRs versus fat tissue (TBR_SUVmax_, TBR_SUVmean_); all the other classes showed increasing TBRs versus fat tissue over time as well.

### 3.6. Inter-Tracer Comparison of Uptake and TBRs Regarding Specific Malignant, Inflammatory and Degenerative Lesions over Time

Appendix A show an inter-tracer comparison between the three tracer variants regarding the uptake within the most relevant subclasses of malignant, thus pancreatic carcinoma of inflammatory/reactive, i.e., inflammatory lesions of the pancreas and of degenerative lesions, ergo degenerative lesions located at acromioclavicular joints or sternoclavicular joints. Inter-tracer differences occurred not only with respect to the absolute values of uptake between FAPI-02, FAPI-46 and FAPI-74 but also due to their curve progression over time as described in the Appendix A.

## 4. Discussion

### 4.1. Summary of the Results

In this retrospective analysis, we demonstrated the differential uptake behavior of several subclasses of malignant-regarding primaries, local recurrences and different localizations of metastases of the inflammatory/reactive with respect to postoperative lesions as well as inflammatory lesions of the pancreas and degenerative lesions of multiple joints and insertion-related tendinopathies over time, adding benefit with respect to distinguishing different types of lesions in FAPI-PET/CT. These differences over time were particularly observed in the direct comparison between the uptakes associated with pancreatic carcinoma and inflammatory lesions of the pancreas. Furthermore, marked differences were also found between the three tracer variants regarding their time-dependent uptake and TBRs versus blood and fat tissue within different subclasses of malignant, inflammatory/reactive and degenerative pathologies.

### 4.2. Differentiation of Malignant from Benign Lesions and the Diagnostic Benefit through Repetitive Early FAPI-PET/CT Imaging

Distinguishing between malignant and benign lesions is one of the big challenges of diagnostic imaging modalities, which is also the case for FAPI-PET/CT imaging. Several studies addressed this issue for FAPI-PET/CT imaging. Hotta et al. discussed within a review article the incidental findings of FAPI PET uptake in various non-oncologic conditions, such as benign tumors, fibrotic, granulomatosis, scarring/wounds, degenerative, as well as inflammatory diseases [17]. They drew the conclusion that interpreting incidental FAPI uptake could be challenging in cancer patients as the uptake was not exclusively seen in malignant but also in benign lesions with a relevant overlap between them. Thus, it was mentioned that the knowledge of physiological and non-oncologic uptake could be helpful to perform accurate interpretation of FAPI-PET uptake. Moreover, Kessler at al. described non-tumor-specific ^68^Ga-FAPI uptake in the majority of patients, i.e., 83.1%, most frequently in degenerative lesions associated with joints and vertebral bones with no significant difference between the two tracer variants ^68^Ga-FAPI-04 and ^68^Ga-FAPI-46 but also within muscles, head and neck localizations and scarring [16]. According to them, the unspecific and nontarget uptake could be generally observed in degenerative, traumatic, inflammatory and physiologic processes at a variety of locations.

In our previous analysis, it was observed that FAPI-46 manifested the highest absolute values of uptake over time compared with FAPI-02 and FAPI-74 [21]. Moreover, it was shown for the three different tracer variants FAPI-02, FAPI-46 and FAPI-74 that malignant, inflammatory/reactive and degenerative pathologies manifested different trends of absolute values of FAPI-PET uptake and of the uptake progression over time by using a repetitive early FAPI-PET/CT imaging with five imaging acquisition timepoints at 10 min, 22 min, 34 min, 46 min and 58 min p.i. [21]. Additionally, within this present study, differences in the uptake behavior were also found for several subclasses of malignant, inflammatory/reactive and degenerative lesions for the three tracer variants on the basis of repetitive early FAPI-PET/CT imaging. Thus, multiple timepoint imaging could be helpful for future imaging protocols to differentiate malignant from benign lesions and to add diagnostic benefits. 

As it was already described in our previous study regarding the selection of the optimal tracer variant depending on the clinical setting and the tissue of interest [21], the inter-tracer-comparison of this study might underline the importance of a careful tracer variant selection, depending on the tissue of interest but also on the suspected type of pathology.

### 4.3. Differentiation of Pancreatic Carcinoma from Inflammatory Lesions of the Pancreas

Since pancreatic carcinoma is considered to be one of the most lethal cancers, distinguishing between malignant and benign lesions of the pancreas is of great importance, for example, for a staging of a suspected local recurrence for which a relapse has to be differentiated from benign inflammatory lesion of the pancreas. Hotta et al. described a physiologically mild to moderate uptake within the pancreas in general and a diffuse pancreatic FAPI uptake in acute pancreatitis that might conceal the PET signaling of malignant lesions in the pancreas [17]. They also discussed non-oncologic causes of focal FAPI uptake with the potential to imitate malignancy.

Our group showed in a previous retrospective study based on 19 patients that PDAC featured a differential trend for uptake kinetics relative to pancreatitis by using multiple timepoint imaging (10 min, 60 min and 180 min) in a way that uptake within tumor lesions tended to remain constant until 60 min p.i. followed by a slight decrease of uptake until the timepoint of 180 min p.i., whereas the pancreatitis-related lesions showed a decrease of uptake from 10 min to 180 min p.i. [10].

This observation is supported by unpublished data of our group with respect to a dual timepoint analysis of 33 patients with suspected recurrent PDAC by applying imaging acquisition timepoints at 20 min and 60 min p.i., which showed a significant decrease of SUV parameters and TBRs over time for inflammatory lesions of the pancreas, while metastatic lesions and local recurrences manifested a relatively stable uptake over time. Thus, dual timepoint imaging improved the diagnostic accuracy for the differentiation between inflammatory and malignant lesions.

By comparing the diagnostic performance of ^68^Ga-FAPI-PET/CT with ^18^F-FDG-PET/CT in relation to radiotracer uptake, diagnostic performance and TNM classifications within the context of suspected and diagnosed pancreatic malignancies, Pang et al. showed not only higher sensitivity in detecting primary pancreatic tumors by using FAPI-PET/CT and that FAPI-PET/CT imaging is superior in terms of TNM but also a stable FAPI uptake over time in pancreatic tumors from 60 min to 180 min p.i., whereas the FAPI uptake linked to pancreatitis decreased over time in the corresponding time interval [24].

The results of our study are in line with the conclusions of the studies mentioned above as malignant and inflammatory lesions of the pancreas manifested differential uptake kinetics from each other, showing a tendency of increase (SUVmax: FAPI-02, (FAPI-46); SUVmean: FAPI-02, FAPI-46) or constant level (SUVmax: (FAPI-46), FAPI-74; SUVmean: FAPI-74) for malignant and of constant level (SUVmean: FAPI-02, FAPI-46) or of decrease (SUVmax: FAPI-02, FAPI-46, FAPI-74; SUVmean: FAPI-74) for inflammatory lesions of the pancreas for the three tracer variants. Thus, the repetitive early FAPI-PET/CT imaging with multiple timepoints of image acquisition may be supportive to differentiate malignant from inflammatory lesions of the pancreas based on the uptake kinetics and added diagnostic value. Multiple timepoint imaging might be useful for future FAPI imaging protocols of the pancreas and beyond.

### 4.4. Limitations

For this retrospective analysis, some limitations have to be taken into consideration. Firstly, due to the relatively low number of 24 patients included, conclusions based on this data set should be drawn with caution. Secondly, another limitation arises from the absence of histological confirmations of the considered pathologies, which were classified only on the basis of clinical information and CT-morphological anomalies. Thirdly, the last acquisition timepoint of this repetitive imaging technique was used to define VOIs for the biodistribution analysis and for pathologies, which were subsequently and automatically transferred to the previous timepoints to generate identical intra-individual VOIs. Although datapoints with visually marked spatial differences as a consequence of movement artefacts within the first four timepoints were excluded from the analysis, this type of artefact can nevertheless lead to an uncertainty for the uptake analysis. Moreover, the surprising observation of markedly increasing uptake over time for degenerative lesions using FAPI-74 should be interpreted with caution due to the small number of degenerative lesions in the corresponding patient cohort. Furthermore, another limitation is occasioned by the heterogeneity of the patient cohort per tracer variant as rarer malignancies were also included in this study. This might have reduced the inter-group comparability although the analysis featured a focus on pancreatic carcinoma. Since no marked differences between the tumor entities were observed in this study, our conclusions could also be relevant beyond the considered tumor entities.

## 5. Conclusions

By applying FAPI-02, FAPI-46 or FAPI-74, repetitive early FAPI-PET/CT imaging manifested differential uptake kinetics and TBRs versus blood and fat tissue over time for several subclasses of malignant, inflammatory/reactive and degenerative lesions, which added diagnostic value for the discrimination of malignant and benign FAPI-PET positive lesions. Particularly with respect to pancreatic carcinoma and pancreatitis, multiple timepoint imaging acquisition showed differential uptake behavior over time for the tracer variants between the malignant and non-malignant pancreatic lesions, which may be helpful for future FAPI imaging protocols. The three tracer variants differ from each other by absolute uptake values and their behavior over time, so tracer selection could be relevant for a specific clinical setting and pathology.

Overall, FAPI-PET/CT imaging in general and multiple timepoint imaging in particular are promising innovative imaging modalities for distinct malignant and benign conditions.

## Figures and Tables

**Figure 1 cancers-14-05301-f001:**
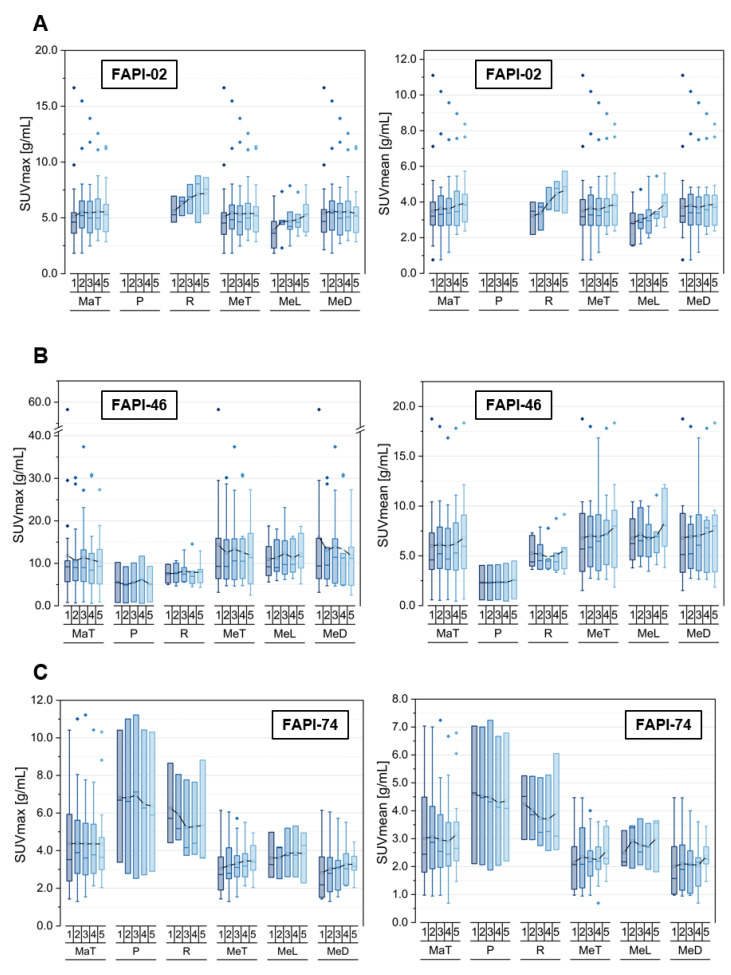
SUVmax and SUVmean values of pooled malignant pathologies (MaT), including primaries (P), local recurrences (R), pooled metastases (MeT) with lymphogenic (MeL) and distant metastases (MeD), over time at 10 min (1), 22 min (2), 34 min (3), 46 min (4) and 58 min (5) after injection of ^68^Ga-FAPI tracer of either FAPI-02 (**A**), FAPI-46 (**B**) or FAPI-74 (**C**). Boxes represent the interquartile range (IQR), whiskers the range of 1.5 IQR, horizontal line within the box indicates the median and small box the mean. Data outliers are shown separately within graph. Trending lines regarding mean are shown.

**Figure 2 cancers-14-05301-f002:**
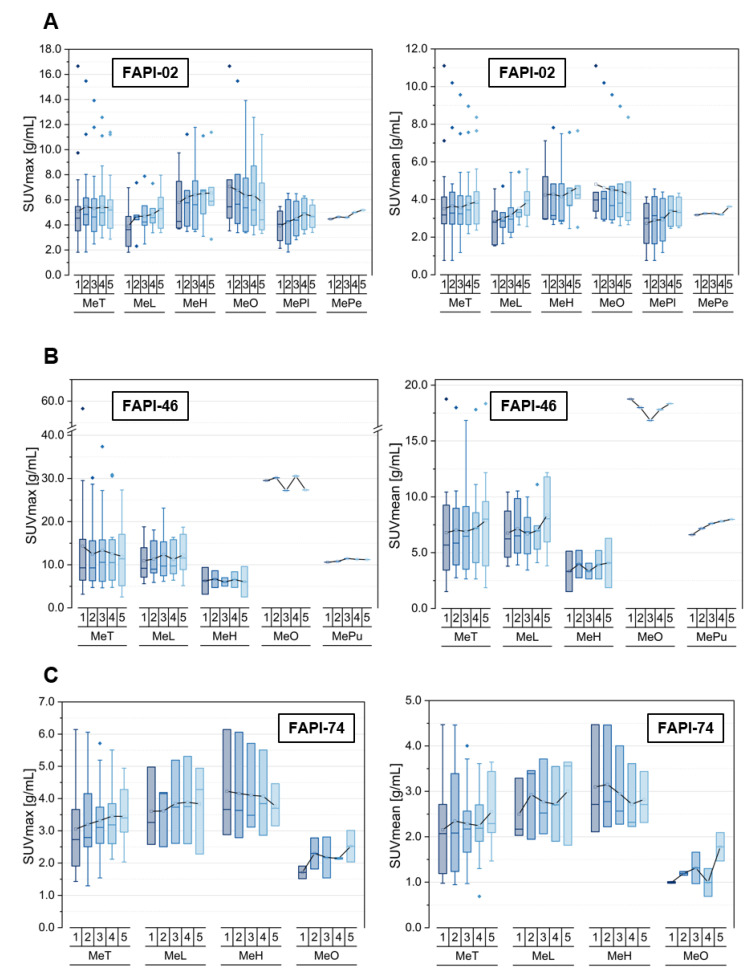
SUVmax and SUVmean values of pooled metastases (MeT) with lymphogenic (MeL) and distant metastases (MeD), including hepatic (MeH), osseous (MeO), pleural (MePl), peritoneal (MePe) and pulmonary metastases (MePu), over time at 10 min (1), 22 min (2), 34 min (3), 46 min (4) and 58 min (5) after injection of ^68^Ga-FAPI tracer of either FAPI-02 (**A**), FAPI-46 (**B**) or FAPI-74 (**C**). Boxes represent the interquartile range (IQR), whiskers the range of 1.5 IQR, horizontal line within the box indicates the median and small box the mean. Data outliers are shown separately within graph. Trending lines regarding mean are shown.

**Figure 3 cancers-14-05301-f003:**
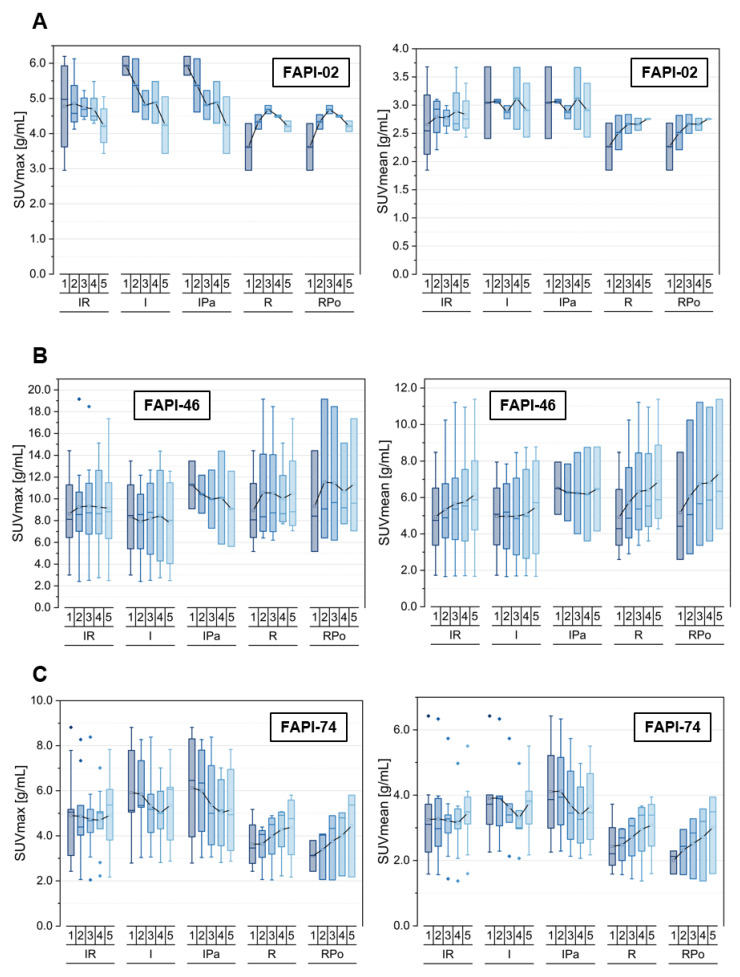
SUVmax and SUVmean values of pooled inflammatory/reactive lesions (IR), including inflammatory manifestations (I) with inflammatory lesions of the pancreas (IPa) and reactive manifestations (R) with postoperative lesions (RPo), over time at 10 min (1), 22 min (2), 34 min (3), 46 min (4) and 58 min (5) after injection of ^68^Ga-FAPI tracer of either FAPI-02 (**A**), FAPI-46 (**B**) or FAPI-74 (**C**). Boxes represent the interquartile range (IQR), whiskers the range of 1.5 IQR, horizontal line within the box indicates the median and small box the mean. Data outliers are shown separately within graph. Trending lines regarding mean are shown.

**Figure 4 cancers-14-05301-f004:**
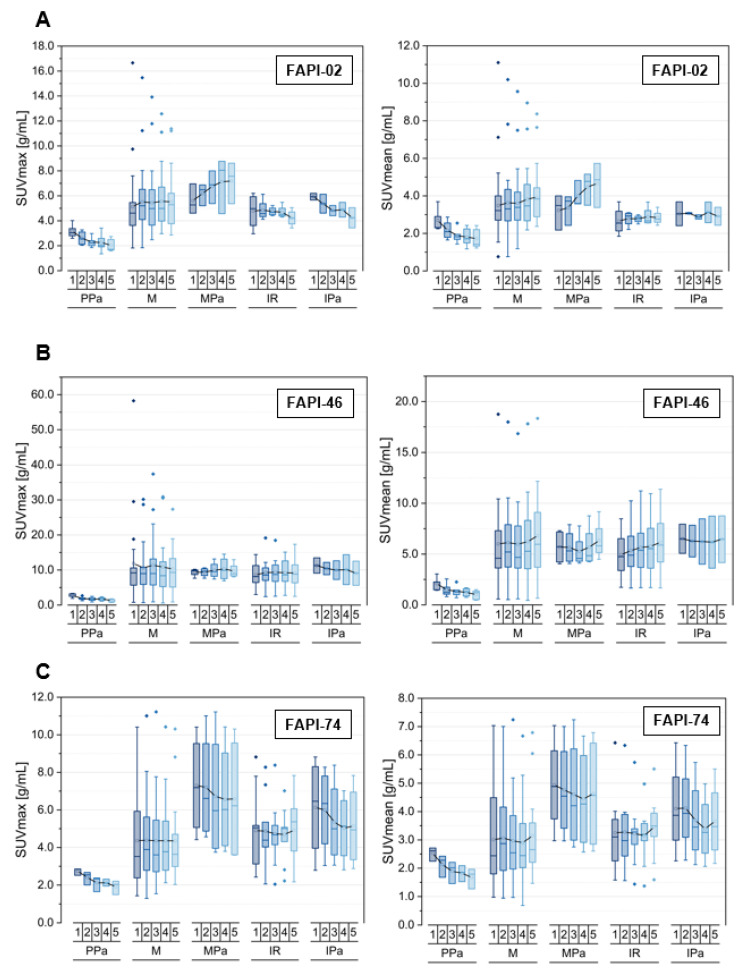
SUVmax and SUVmean values of the pancreas, including physiological pancreas tissue (PPa), pooled malignant lesions (M) with malignant lesions of pancreas (MPa) and pooled inflammatory/reactive lesions (IR) with inflammatory lesions of pancreas (IPa), over time at 10 min (1), 22 min (2), 34 min (3), 46 min (4) and 58 min (5) after injection of ^68^Ga-FAPI tracer of either FAPI-02 (**A**), FAPI-46 (**B**) or FAPI-74 (**C**). Boxes represent the interquartile range (IQR), whiskers the range of 1.5 IQR, horizontal line within the box indicates the median and small box the mean. Data outliers are shown separately within graph. Trending lines regarding mean are shown.

**Figure 5 cancers-14-05301-f005:**
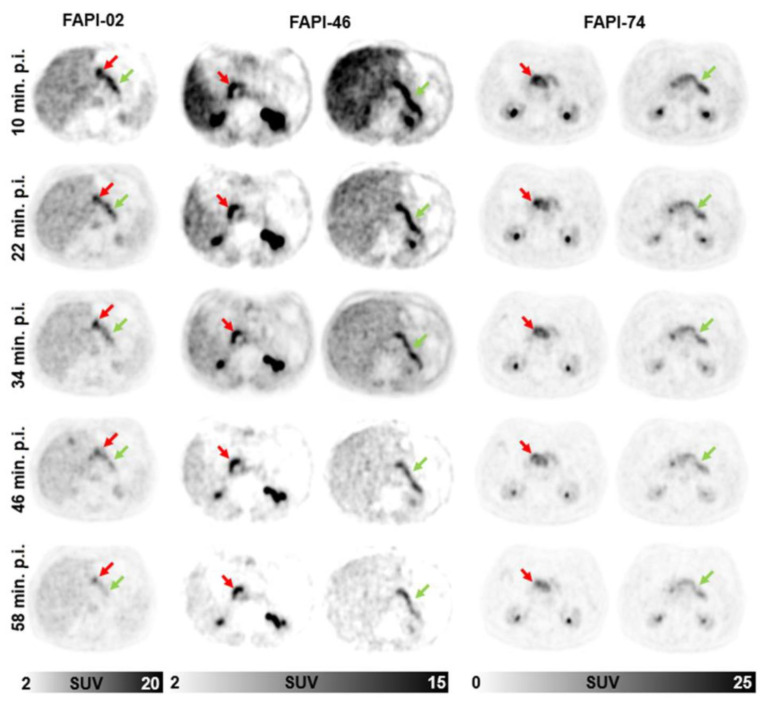
Representative axial PET images of pancreatic carcinoma- (red arrow) and pancreatitis- (green arrow) associated uptake by ^68^Ga-FAPI-PET/CT imaging for FAPI-02 (left column, 67 year old male patient with resected pancreatic carcinoma, staging with suspected local recurrence, local recurrence, six metastases), FAPI-46 (middle column, 64 year old male patient with resected pancreatic cancer, staging with suspected local recurrence, no metastases) or FAPI-74 (right column, 60 year old male patient with pancreatic carcinoma, staging in advance of radiation therapy, primary, single hepatic metastasis) over time with imaging acquisition timepoints 10 min (1), 22 min (2), 34 min (3), 46 min (4) and 58 min (5) after application.

**Figure 6 cancers-14-05301-f006:**
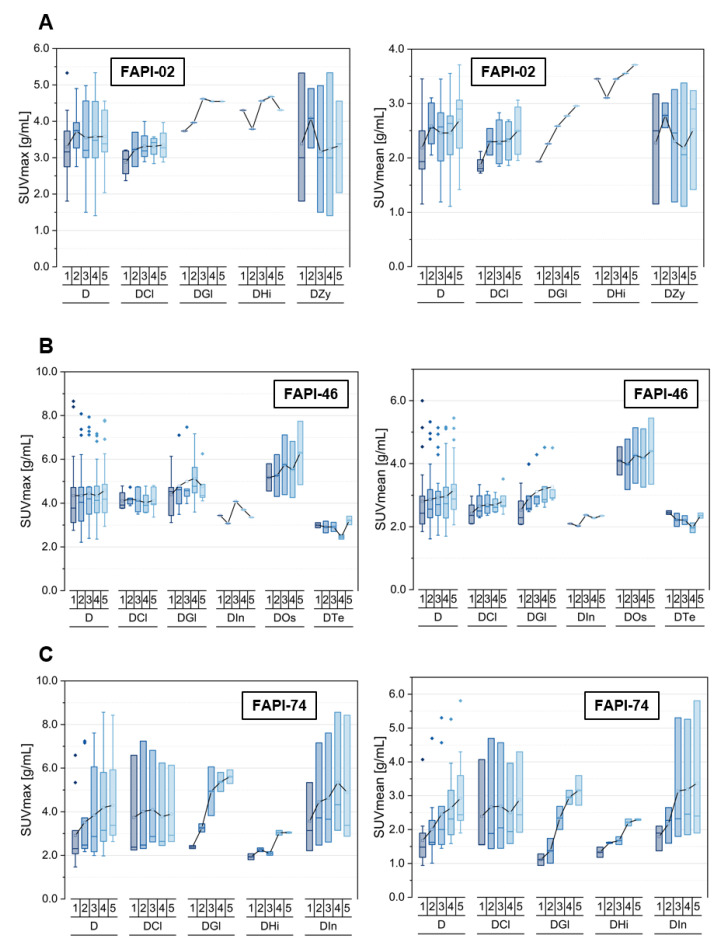
SUVmax and SUVmean values of degenerative lesions (D) located at acromioclavicular joints or sternoclavicular joints (DCl), at zygapophysial joints (DZy), glenohumeral joints (DGl), hip joints (DHi), osteophytes (DOs), temporomandibular joints (DTe) and including insertion-related tendinopathy (DIn; located at collum femoris) over time at 10 min (1), 22 min (2), 34 min (3), 46 min (4) and 58 min (5) after injection of ^68^Ga-FAPI tracer of either FAPI-02 (**A**), FAPI-46 (**B**) or FAPI-74 (**C**). Boxes represent the interquartile range (IQR), whiskers the range of 1.5 IQR, horizontal line within the box indicates the median and small box the mean. Data outliers are shown separately within graph. Trending lines regarding mean are shown.

**Table 1 cancers-14-05301-t001:** Number of malignant pathologies (M), i.e., primaries (P), local recurrences (R), total metastases (MeT) with lymphogenic (MeL) and distant metastases (MeD), including hepatic (MeH), osseous (MeO), pleural (MePl), peritoneal (MePe) and pulmonary metastases (MePu), for each radiotracer variant.

Tracer Variant	Pathology				Number (%)			
FAPI-02	Total				34 (69.4)			
		P				0 (0.0)		
		R				3 (8.8)		
		MeT				31 (91.2)		
			MeL				7 (22.6)	
			MeD				24 (77.4)	
				MeH				5 (20.8)
				MeO				7 (29.2)
				MePl				4 (16.7)
				MePe				1 (4.2)
				other				7 (29.2)
FAPI-46	Total				21 (40.4)			
		P				2 (9.5)		
		R				5 (23.8)		
		MeT				14 (66.7)		
			MeL				5 (35.7)	
			MeD				9 (64.3)	
				MeH				2 (22.2)
				MeO				1 (11.1)
				MePu				1 (11.1)
				other				5 (55.6)
FAPI-74	Total				16 (44.4)			
		P				3 (18.8)		
		R				3 (18.8)		
		MeT				10 (62.5)		
			MeL				3 (30.0)	
			MeD				7 (70.0)	
				MeH				3 (42.9)
				MeO				2 (28.6)
				other				2 (28.6)

**Table 2 cancers-14-05301-t002:** Number of inflammatory/reactive lesions (IR), i.e., inflammatory (I) with inflammatory lesions of pancreas (IPa) and inflammatory lesions of esophagus (IOe), reactive (R) with postoperative lesions (RPo) for each radiotracer variant.

Tracer Variant	Pathology			Number (%)		
FAPI-02	Total			4 (100.0)		
		I			2 (50.0)	
			IPa			2 (100.0)
			IOe *			0 (0.0)
			Other			0 (0.0)
		R			2 (50.0)	
			RPo			2 (100.0)
			Other			0 (0.0)
FAPI-46	Total			8 (100.0)		
		I			4 (50.0)	
			IPa			2 (50.0)
			IOe *			2 (50.0)
			Other			
		R			4 (50.0)	
			RPo			3 (75.0)
			Other			1 (25.0)
FAPI-74	Total			9 (100.0)		
		I			5 (55.6)	
			IPa			4 (80.0)
			IOe *			0 (0.0)
			Other			1 (20.0)
		R			4 (44.4)	
			RPo			3 (75.0)
			Other			1 (25.0)

* not visualized in Figure 3 and the Appendix A (s. Appendix A).

**Table 3 cancers-14-05301-t003:** Number of physiological pancreas tissues (PPa), of pooled malignant lesions (M) with malignant lesions of pancreas (MPa), of pooled inflammatory/reactive lesions (IR) with inflammatory lesions of pancreas (IPa) for each radiotracer variant.

Tracer Variant	Pathology		Number (%)	
FAPI-02	PPA		5	
	M		34 (100.0)	
		MPa		3 (8.8)
	IR		4 (100.0)	
		IPa		2 (50.0)
FAPI-46	PPA		5	
	M		21 (100.0)	
		MPa		4 (19.0)
	IR		8 (100.0)	
		IPa		2 (25.0)
FAPI-74	PPA		3	
	M		16 (100.0)	
		MPa		4 (25.0)
	IR		9 (100.0)	
		IPa		4 (44.4)

**Table 4 cancers-14-05301-t004:** Number of degenerative lesions (D) located at acromioclavicular joint or sternoclavicular joint (DCl)**,** at zygapophysial joint (DZy), glenohumeral joint (DGl), hip joint (DHi), osteophyte (DOs), temporomandibular joint (DTe), insertion-related tendinopathy (DIn, localized at collum femoris) and at spinous process (DSp) for each radiotracer variant.

Tracer Variant	Pathology		Number (%)	
FAPI-02	Total		9 (100.0)	
		DCl		4 (44.4)
		DZy		3 (33.3)
		DGl		1 (11.1)
		DHi		1 (11.1)
		Dos *		0 (0.0)
		DTe *		0 (0.0)
		Din *		0 (0.0)
		DSp *		0 (0.0)
		Other *		0 (0.0)
FAPI-46	Total		22 (100.0)	
		DCl		6 (27.3)
		DZy		1 (4.5)
		DGl		5 (22.7)
		DHi *		0 (0.0)
		DOs		2 (9.1)
		DTe		2 (9.1)
		DIn		1 (4.5)
		DSp *		2 (9.1)
		Other *		3 (13.6)
FAPI-74	Total		11 (100.0)	
		DCl		3 (27.3)
		DZy *		0 (0.0)
		DGl		2 (18.2)
		DHi		2 (18.2)
		Dos *		0 (0.0)
		DTe *		0 (0.0)
		DIn		3 (27.3)
		DSp *		1 (9.1)
		Other *		0 (0.0)

* not visualized in Figure 6 and the Appendix A (s. Appendix A).

## Data Availability

The data presented in this study are available on request from the corresponding author.

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
