# Peer review of "Subclass Analysis of Malignant, Inflammatory and Degenerative Pathologies Based on Multiple Timepoint FAPI-PET Acquisitions Using FAPI-02, FAPI-46 and FAPI-74"

_cancers, 2022, doi:10.3390/cancers14215301_

Round 1

Reviewer 1 Report

The direction of this research is very novel and has great academic significance. However, there is a limitation that PET tracer is not a single type. Also, it is difficult to understand the difference between a malignant lesion and a benign lesion from this paper alone.

Therefore, I think that it will be a very meaningful research result if the entire analysis target is classified into malignant lesions and benign lesions and presented so that readers can understand the differences between the two groups.

Author Response

Subclass analysis of malignant, inflammatory and degenerative pathologies based on multiple timepoint FAPI-PET acquisitions using FAPI-02, FAPI-46 and FAPI-74

Manuscript number: cancers-1946729

Point by point reply to the reviewer´s comments

Reviewer 1

Comments and Suggestions for Authors

The direction of this research is very novel and has great academic significance. However, there is a limitation that PET tracer is not a single type. Also, it is difficult to understand the difference between a malignant lesion and a benign lesion from this paper alone. Therefore, I think that it will be a very meaningful research result if the entire analysis target is classified into malignant lesions and benign lesions and presented so that readers can understand the differences between the two groups.

  1. Answer: Thank you for your appreciative and constructive general comments. The reason why more than one variant of FAPI radiotracers is considered is that this manuscript provides a subgroup analysis of the dataset of a previous published article of our group comparing the three different tracer variants FAPI-02, FAPI-46 and FAPI-74 in a systematic manner to highlight potential differences between the tracer variants [Glatting FM, Hoppner J, Liew DP, et al. Repetitive early FAPI-PET acquisition comparing FAPI-02, FAPI-46 and FAPI-74: methodological and diagnostic implications for malignant, inflammatory and degenerative lesions. J Nucl Med. 2022.]. Moreover, the aim of this analysis is to focus not only on the general differences between malignant and benign lesion, but to evaluate the differential behaviour of multiple subclasses over time in order to answer the question whether marked differences can be observed for several types of malignant, inflammatory/reactive and degenerative lesions. The general comparison between malignant and benign lesions has already been included in the previous published article and was the basis for the further subgroup analysis that was performed within this research.

Reviewer 2 Report

The manuscript entitled "Subclass analysis of malignant, inflammatory and degenerative pathologies based on multiple timepoint FAPI-PET acquisitions using FAPI-02, FAPI-46 and FAPI-74" is a well-written article which deals with novel radiotracers and aspects of contemporary nuclear medicine. It is based on the additional subgroup analysis of the dataset of 24 cancer patients previously published by the same team. It describes the role of multiple timepoint FAPI-PET/CT imaging in distinguishing benign from malignant entities. It shows new and clinically relevant data, especially regarding the non-invasive diagnosis of pancreatic tumors, which may bring benefit in the field of oncology. The main limitation of the study is a relatively low number of patients and a lack of histopathological verification of the imaged pathologies. I have a few minor concerns listed below. Once the authors include them in the corrected version of the manuscript, I would kindly recommand the editors to accept the article for publication.

Comments:

lines:

51: 'others' should be 'other'

58: 'processes and reactive tissue' should be 'processes, and reactive tissue'

146: 'uptake of over time' should be 'uptake over time' 

217-222 - the sentence is too long, please split it into two shorter ones.

305: 'that not shown' should be 'that are not shown'

404: 'adding benefit with respect to distinguishing different types of lesions' 

should be 'add benefit with respect to distinguishing different types of lesions in FAPI-PET/CT'

492: 'VOIS' should be 'VOIs'

The sentence in Supplemental Materials 'FAPI-46 showed the highest absolute values of uptake over time compared with FAPI-02 and FAPI-74.' should be placed in the Discussion or even the Conclusions in the main part of the manuscript, since it is on of the important observations of this study. The differences of uptake levels were manifested in all the comparisons.

Author Response

Subclass analysis of malignant, inflammatory and degenerative pathologies based on multiple timepoint FAPI-PET acquisitions using FAPI-02, FAPI-46 and FAPI-74

Manuscript number: cancers-1946729

Point by point reply to the reviewer´s comments

Reviewer 2

Comments and Suggestions for Authors

The manuscript entitled "Subclass analysis of malignant, inflammatory and degenerative pathologies based on multiple timepoint FAPI-PET acquisitions using FAPI-02, FAPI-46 and FAPI-74" is a well-written article which deals with novel radiotracers and aspects of contemporary nuclear medicine. It is based on the additional subgroup analysis of the dataset of 24 cancer patients previously published by the same team. It describes the role of multiple timepoint FAPI-PET/CT imaging in distinguishing benign from malignant entities. It shows new and clinically relevant data, especially regarding the non-invasive diagnosis of pancreatic tumors, which may bring benefit in the field of oncology. The main limitation of the study is a relatively low number of patients and a lack of histopathological verification of the imaged pathologies. I have a few minor concerns listed below. Once the authors include them in the corrected version of the manuscript, I would kindly recommand the editors to accept the article for publication.

Answer: Thank you for your appreciative and constructive general comments.

Comments:

Lines:

57: 'others' should be 'other'

Answer: Thank you for this advice. We changed “others” into “other”.

58: 'processes and reactive tissue' should be 'processes, and reactive tissue'

Answer: Thank you for this advice. We fully agree and changed “processes and reactive tissue” into “processes, and reactive tissue”.

146: 'uptake of over time' should be 'uptake over time' 

Answer: Thank you for this advice. We changed “uptake of over time” into “uptake over time”.

217-222 - the sentence is too long, please split it into two shorter ones.

Answer: Thank you for your comment. We fully agree to split the corresponding sentence into two short ones as follows:

 “Respecting FAPI-46, the pooled inflammatory and reactive lesions showed a constant level of uptake (SUVmax) or a slightly increasing uptake (SUVmean) over time. These trends were made up by decreasing SUVmax values and more or less constant SUVmean values of general inflammatory and pancreatitis-associated lesions in combination with increasing uptake values in terms of SUVmax and SUVmean concerning reactive, particularly postoperative, lesions.“

305: 'that not shown' should be 'that are not shown'

Answer: Thank you for this advice. We changed “that not shown” into “that are not shown”.

404: 'adding benefit with respect to distinguishing different types of lesions' 

should be 'add benefit with respect to distinguishing different types of lesions in FAPI-PET/CT'

Answer: Thank you for this advice. We fully agree and added the information “in FAPI-PET/CT” as suggested.

492: 'VOIS' should be 'VOIs'

Answer: Thank you for this advice. We changed “VOIS” into “VOIs”.

The sentence in Supplemental Materials 'FAPI-46 showed the highest absolute values of uptake over time compared with FAPI-02 and FAPI-74.' should be placed in the Discussion or even the Conclusions in the main part of the manuscript, since it is one of the important observations of this study. The differences of uptake levels were manifested in all the comparisons.

Answer: Thank you for this comment. As this observation has already been made in the previous analysis of this dataset, this sentence was added in the discussion of the main part in line 429 as follows:

In our previous analysis, it was observed that FAPI-46 manifested the highest absolute values of uptake over time compared with FAPI-02 and FAPI-74 [21]. Moreover, it was shown for the three different tracer variants FAPI-02, FAPI-46 and FAPI-74 that malignant, inflammatory/reactive and degenerative pathologies manifested different trends of absolute values of FAPI-PET uptake and of the uptake progression over time by using a repetitive early FAPI-PET/CT imaging with five imaging acquisition timepoints at 10 min, 22 min, 34 min, 46 min and 58 min p.i.[21].”

Round 2

Reviewer 1 Report

Unfortunately, the comments I made last time on this paper do not seem to have been fully reflected. Accordingly, my opinion remains unchanged.